# A more representative "best representative value" for daily total column ozone reporting

Andrew R.D. Smedley[1], John S. Rimmer[1], Ann R. Webb[1]

Centre for Atmospheric Science, University of Manchester, Manchester, M13 9PL, United Kingdom

*Correspondence to*: Andrew R.D. Smedley (andrew.smedley@manchester.ac.uk)

**Abstract.** Long-term trends of total column ozone, assessments of stratospheric ozone recovery and satellite validation are underpinned by a reliance on daily "best representative values" from Brewer spectrophotometers and other ground-based ozone instruments. In turn reporting of these daily total column ozone values to the World Ozone and Ultraviolet Data Centre has traditionally been predicated upon a simple choice between direct sun and zenith sky observations. For mid- and high-latitude
monitoring sites impacted by cloud cover we discuss the potential deficiencies of this approach in terms of its rejection of otherwise valid observations and capability to evenly sample throughout the day. A new methodology is proposed that makes full use of all valid direct sun and zenith sky observations, accounting for unevenly spaced observations and their relative uncertainty, to calculate an improved estimate of the daily mean total column ozone. It is demonstrated that this method can increase the number of contributing observations by a factor of 2.5, increases the sampled time span, and reduces the spread
of the representative time by half. The largest improvements in the daily mean estimate are seen on days with the smallest number of contributing direct sun observations. No effect on longer-term trends is detected, though for the sample data analysed we observe a mean increase of 2.8 DU (0.82 %) w.r.t. the traditional direct sun vs zenith sky average choice. To complement the new calculation of a best representative value of total column ozone and separate its uncertainty from the spread of observations, we also propose reporting its standard error rather than the standard deviation, together with measures
of the full range of values observed.

## 1 Introduction

Global ground-based monitoring of total column ozone (TCO) relies on the international network of Brewer spectrophotometers since they were first developed in the 1980s (Kerr et al., 1981), which has expanded the number of sites and measurement possibilities from their still-operating predecessor instrument, the Dobson spectrophotometer. Together these
networks provide validation of satellite retrieved total column ozone as well as instantaneous point measurements that have value for near real time low ozone alerts, particularly when sited near population centres, as inputs to radiative transfer models at ultraviolet wavelengths, and critically underpin the monitoring requirement of The Vienna Convention for the Protection of the Ozone Layer, 1985.

There are recent indications that for the first time since the treaty was enacted, and CFCs and other depleting substances were banned, the ozone layer is showing signs of recovery (WMO, 2014, and references therein). These and related trend analyses, however, use daily mean TCO values as their starting point. In the parlance of the World Ozone and Ultra-Violet Data Centre (WOUDC), the WMO data centre for ground-based ozone data, the daily values submitted by data originators should be the "best representative values" (WOUDC, 2016). For Brewer spectrophotometers a cascade of choices is recommended as follows. If available, the mean of valid direct sun (DS) measurements is preferred. If no valid DS observations are available for a given day, then the mean of all valid zenith sky (ZS) measurements is used. If no valid DS or ZS observations are available, the last choice is to rely on the mean of valid focused moon observations, a measurement mode predominantly used at high latitude stations. Here we only consider the choice between DS and ZS observations.

The majority of effort spent on calibrating Brewer spectrophotometers is directed towards ensuring high quality DS calibrations are distributed globally through the Brewer reference triad (Fioletov et al., 2005), intercomparisons at the Regional European Calibration Centre and through initiatives such as COST Action ES1207 and EUBrewNet. ZS observations are then linked to the DS calibration through a polynomial fit of quasi-synchronous DS and ZS observations (Kipp & Zonen, 2005). This additional calibration step explains the default preference for DS over ZS measurements as it incurs a small associated uncertainty. However at mid- and high-latitude stations in particular the annual mean cloud fraction can exceed 50 % (Wilson and Jetz, 2016) and limits opportunities for recording viable DS observations. As a consequence, for a high fraction of days the best representative daily value (BRDV) is based upon zenith sky measurements. More crucially during partly cloudy days, the BRDV can be reliant on a small number of individual DS observations (<5), which may be biased towards either the start or end of the day, whilst a greater number of valid ZS observations from throughout the observation period are rejected

This gives rise to the question: could a more representative daily value be obtained from an increased number of ZS measurements than from a small number of DS measurements? To answer this still forces a choice between valid DS and ZS observations as the number of DS measurements falls – whichever set of observations is chosen, a set of otherwise valid data is not incorporated into the calculation of the best representative value. Therefore we propose an alternative methodology to calculate a best daily representative value that retains both direct sun and zenith sky measurements, taking into account their relative uncertainties and periods of time when valid measurements are more frequent.

## 2 Instrument and data processing description

Brewer spectrophotometers, their operation and standard data processing routes have been described previously in the literature (Brewer, 1973; Fioletov et al., 2005; Smedley et al., 2012; Savastiouk and McElroy, 2005). For context we outline the key points here. The core of each instrument is a single or double monochromator unit whose output is detected by a photomultiplier tube. For the DS measurement mode the input is from a rotating prism assembly pointed towards the sun's disc. Column ozone

observations are achieved by rapidly repeated measurements at 5 operational wavelengths over a period of approximately 3 minutes, and a final value calculated by implementing the Lambert-Beer law and knowledge of the absorption cross-section of ozone molecules. ZS observations are made in the same way but the rotating prism is instead directed to collect scattered light from the zenith, and then an empirical polynomial adjustment is applied. This polynomial adjustment assumes that the apparent ozone column from the zenith sky measurement is quadratic in both air mass factor and the actual column ozone. The nine constants necessary are determined from a large number of quasi-simultaneous DS and ZS measurements (>500) and are instrument and site specific. This relationship is usually determined at the instrument's home site, rather than during an intercomparison or calibration exercise, though Fioletov et al (2011) described an improved RT modelling based methodology that reduced the instrument-specific unknowns to two parameters (though nine constants are still necessary). However the polynomial constants are determined, the ZS observation is then found by solving the relevant quadratic equation.

## 3 A more representative daily value

For a site where direct sun can be guaranteed for the majority of the day, an instrument could be scheduled to only attempt DS observations at regular intervals (together with the necessary diagnostic routines) and the mean of these observations will be a reliable estimate of the actual daily mean TCO overhead at the station. For other sites where cloud is more variable and unpredictable, the observational schedule must contain a combination of both ZS and DS measurements. However local cloud cover conditions may only permit a small number of DS measurements to be successfully recorded. Figure 1 shows an example day where only 4 DS observations were recorded between 13:48 and 15:49 UTC, and where their arithmetic mean (298.23 DU) differs substantially from the daily mean TCO at the station as indicated by ZS observations. In order to avoid the potential binomial choice between a small number of DS observations and a greater number of ZS observations, we propose a weighted daily mean that utilises all valid DS and ZS values.

Our aim is to construct a daily mean that has the following properties. In the absence of either any valid DS or ZS observations it produces the same result as the standard method (once clustering of observations is accounted for). With the addition or subtraction of a single DS data point, there is a graceful change in the BRDV and the overall time period it represents. It should represent as fully as possible the day's TCO observations. It should give equal weighting to equal periods of time and hence account for time clustering of valid observations and for their relative uncertainties. It should be able to be applied to historic data and not necessitate any changes to the instrument's future schedule or data collection routines.

The proposed methodology is as follows. The first pre-requisite is for ZS polynomials to be assessed regularly (here these have been recalculated using the full available dataset for each inter-calibration period) to ensure the individual DS and ZS observations are comparable and there is minimal bias in the ZS observations (for example in our dataset overall DS–ZS bias = –0.4 DU; standard deviation of distribution of individual DS–ZS pairs = 6.3 DU).

All DS and ZS measurements are then filtered to remove those that do not meet the validity criteria. Observations that have a standard deviation of > 2.5 DU for DS and > 4.0 DU for ZS are rejected. We note that the standard choice of standard deviation threshold is 2.5 DU for ZS observations, but increasing the limit to 4 DU does not introduce any bias and increases the total number of valid observations (Fioletov et al., 2005; Fioletov et al., 2011). Observations at air mass factors > 4 are also rejected for single monochromator instruments, but this limit is raised to 6 for double monochromator instruments due to their improved stray light rejection (Karppinen et al., 2015). To ensure that any residual bias is not present at high air mass factors, an additional tail removal step is applied. For this the day's data is smoothed with a 30 minute running average filter, and end periods of time where the smoothed TCO exhibits apparent rates of change > 20 DU/hr are identified. Any observations falling within these periods are then removed.

At this stage the remaining DS and ZS values meet the specified validity criteria, and have passed the additional tail removal check. To form a BRDV from these individual observations, we calculate a weighted mean of the full set of data points, but where the weighting has two components: the time for which the observation is representative, and the uncertainty of each observation, as in Eq. (1):

$$\overline{X} = \frac{\sum_i X_i w_i}{\sum_i w_i} \tag{1}$$

where $\overline{X}$ is the BRDV, $X_i$ are the individual observations (both DS and ZS), and $w_i$ is the weighting for each observation. The weighting is defined in Eq. (2) as:

$$w_i = \frac{t_i}{\sigma_i^2}, \tag{2}$$

where $t_i$ is the time from the midpoint of the preceding inter-observation time interval to the midpoint of the following inter-observation time interval. For the first data point we instead use the length of the first inter-observation time period, and likewise for the last data point. In all cases $\sigma_i$ are the uncertainties for each individual observation, taken as the normal measurement standard deviation and used as part of the validity test. If no account were taken of the relative uncertainties of each observation, nor of their time intervals, this formulation reduces to the simple arithmetic mean of the valid observations.

We also note that this methodology could be applied to all data acquired without applying a threshold standard deviation validity filter as data points with large errors will contribute to the BRDV proportionally less. However more care needs to be taken as regards relaxing the airmass threshold requirement as small biases may be introduced, inflated by the effect of observing at high airmass, whilst the uncertainty will not have been captured by the intrinsic standard deviation of the observation. Further the ZS uncertainties could be expanded appropriately to account for any day-to-day bias between ZS and

DS observations under differing sky conditions, or alternatively to incorporate the DS-ZS polynomial fit mean residual, for example.

For higher latitude sites where other measurement modes are relied upon, such as focussed moon, focussed sun, or TCO derived from global spectral irradiance, these observations could also be incorporated into the BRDV calculation in a similar way (see seasonal variation of observation types in Karppinen et al, 2016). The prerequisites would be that each individual observation had an associated uncertainty and that the observations from different measurement modes had been homogenised beforehand. In terms of practical implementation, if the method were adopted by the community a new observation type would have to be registered at WOUDC (a mechanism that is already available), with relevant details added to the Scientific Support Statement as necessary. For stations that submit raw data, or processed individual observations, the weighted mean BRDV calculation could be applied across all sites as a daily summary value.

It is also worthwhile considering the strengths of this methodology under specific theoretical conditions. If, for example, on a given day there is a strong linear east-west ozone gradient present, then the most appropriate daily measure should return a value similar to the TCO above the site. The traditional method risks producing a daily value that could be substantially different if, due to cloud cover, only a few valid DS measurements can be recorded during early morning or late in the day when the TCO is being sampled to the west or east of the site. In partly cloudy or cloudy conditions a minimum airmass TCO measurement may not be obtained. In contrast the proposed method guards against these issues as ZS observations could be sampled more fully through the day, whilst the contribution from DS measurements would represent the effective TCO along the slant path when the direct solar beam is visible. As a result the proposed BRDV TCO calculation is more appropriate for UV exposure studies than the traditional calculation, more representative of the conditions throughout the day, and more resilient than relying on a single value at minimum airmass, for example. For non-linear spatial gradients in TCO then limited DS measurements could result in a value more different still from the mean TCO overhead, while the bias from selecting the TCO near minimum airmass would depend on the spatial distribution of ozone.

## 4 Sample results

To demonstrate the impact of this method on real world data, we apply it to the 2000–2016 data record from Brewer spectrophotometer #172, located in Manchester, UK [53.47°N, 2.23°W] (see table 1 and fig 2). For context the minimum airmass observed at this location during the summer solstice is approximately 1.15, whilst during the winter solstice it is 4.15.

Overall we see an increase in the mean number of contributing observations (N) from 10.72 to 24.06, with the upper 10th percentiles also increasing from 24 to 46. The effect is dominated by summer-time measurements that show an increase of 150 % from 14.68 to 36.68 averaged over the three months bracketing the summer solstice. Whilst the effect is still present

during winter months (when data collection is inherently more difficult due to the lower solar elevations), the improvement is smaller: the mean N increasing from 5.71 to 8.54 contributing observations per day.

As expected we see concomitant tightening of the distribution of representative times (defined as the mean of valid observational times, weighted by their TCO values) around solar noon (close to midday). There is also a skewing of the observational time span distribution to longer periods. The representative time is symmetrical about solar noon in both the traditional and new methods, but the width of the annual distribution (defined as the interdecile range) is halved from 4.05 h to 2.01 h, showing the new method results in BRDVs more representative of conditions at solar noon Again due to the longer day length the improvement is accentuated during summer (6.5 h reduced to 2.7 h), but still present during winter months (1.56 h reduced to 1.20 h). Time spans for the whole year are generally skewed to the right-hand side of the distribution, though the upper and lower bounds do not change (being limited by number of daylight hours and instances of single contributing observations respectively). Much of the skewness in the annual distribution is attributable to that occurring during the summer subset (fig 2, third row, second column), where the lower 10th percentile increases from 0.5 h to 10.0 h.

Taken together these results demonstrate that the method enables a more representative daily mean to be calculated, predominantly by sampling more fully through each day and over a wider range of weather conditions. However it is prudent to investigate the impact on the overall time series and trends.

Focussing on the 2006–2016 subset for clarity, in figure 3 we see only a small impact on the monthly mean TCO values from applying the methodology described, with no discernible trend or annual cycle in the difference (fig. 3, upper and middle panels). Regression analysis shows the trend in the difference between traditional and proposed methodologies to be –0.0153 DU/yr, but this is not significant (Kendall-Mann test, $p = 0.275$). Specifically over the 2006 to 2016 period we find a skewed distribution of daily differences between the new and traditional BRDVs with a mean increase of 2.79 DU, an amount comparable to the calibration uncertainty (the spread of values exhibited by different instruments immediately after calibration), and with the upper and lower 10th percentiles being –2.67 DU and 10.92 DU respectively (equivalent to –0.79 % and +3.22 % of the annual mean TCO). The methodology does not substantially alter the form of the time series with an $R^2$ coefficient of 0.984 (fig 3, lower middle panel). It should be emphasized that this bias of 2.79DU does not result from the ZS polynomial procedure, nor is it related to any bias between individual DS and ZS measurements: the overall bias of the polynomial fit is only –0.4DU. Instead the bias noted here relates to the new method's increased sampling over longer daylight hours. The underlying cause is likely related to a longer sampling period where the TCO is larger, or to increased reliance on different internal filters. We anticipate that the former is dominant as figure 3 (middle panel) suggests increased differences during summer months when the time span is increased the most.

In figure 3 (lower right panel) we explore the ranges of differences between methodologies further. It is anticipated that the greatest differences between traditional and proposed methods will be seen on days with high ozone variability and a low number of contributing DS measurements (or a high fraction of ZS observations). Plotting the TCO difference against the number of valid DS measurements, the greatest variability is seen for a single valid DS measurement with the distribution rapidly narrowing as the number of valid DS observations increases. We distinguish days according to their seasons, with those from summer and autumn [JJASON] marked in dark grey and winter and spring [DJFMAM] marked in red. Typically TCO exhibits a much larger variability during winter and spring at this location, though the effect is not overly strong and the number of DS observations is a better indicator for a large potential improvement in the BRDV.

Together these results suggest that there should be no impact on long-term trends at a site where the data record is derived from a single instrument type. However, there could be implications where there has been a change in data sampling method. Moving from a semi-manual Dobson spectrophotometer that makes a limited set of observations on a predefined schedule to a Brewer spectrophotometer that operates quasi-continuously and selects a daily value on the traditional DS vs ZS choice, could introduce a small step change due to this affect, which may contribute to a perceived trend. Likewise applying the proposed method to only part of a data record, because individual historical measurements have been lost, for example, could also introduce a small step in the overall record.

Testing the influence of the new methodology in terms of the agreement between ground-based and satellite retrievals (fig. 4), we find a marginal improvement in the ground vs satellite TCO for daily mean data in terms of their $R^2$ correlation (0.9560 vs 0.9455), and best fit slope (1.0041 vs 0.9930). More noticeably, there is a narrowing of the distribution of differences (ground-minus-satellite) from an interdecile range of 26.86 DU to 24.01 DU, whilst the mean is shifted to higher biases (from 10.49 DU under the traditional DS-ZS preference to 13.28 DU). Satellite retrievals were also compared against the closest individual observation to the overpass time under two assumptions. First, selecting the nearest valid DS measurement as a preference, and if none were available, then selecting the closest ZS observation (equivalent to the traditional DS-ZS choice). Second, selecting the closest observation to the mean overpass time with no preference for observation type (equivalent to the proposed BRDV calculation). The results nonetheless were very similar to those for BRDV in figure 4.

## 5 Measures of daily spread and estimating real time TCO

While the focus of this study is on the determination of a more representative daily TCO value, there are a number of related issues concerning reporting of the daily spread that will be discussed in this section.

At present WOUDC recommendations are to report, in addition to the best representative value, a standard deviation for the day's observations, which implicitly assumes a normal distribution. Whether a day's observations of the underlying ozone

column necessarily falls within a normal distribution is not obvious, nor are the authors aware of any evidence in the literature. To that end we have applied the Kolmogorov-Smirnov test (Massey, 1951) to each day's observations for the same 2006–2016 subset of data. The null hypothesis for this test is that the individual daily observations come from a standard normal distribution and on application we find that this null hypothesis is not rejected for any of the days in our test sample. That is, all can be considered being taken from a normal distribution.

Whilst this result does not undermine the use of the standard deviation as a measure of the spread of the day's data, we propose that other metrics may be more useful. Specifically to separate out the uncertainty in the best representative daily value from the range exhibited by individual observations, a more useful measure would be to use the standard error of the weighted mean to indicate the uncertainty of the best representative value, plus additional metrics relating to the maximum and minimum TCO observed. These latter could be the strict maximum and minimum, or to guard against the influence of short-duration spikes, the upper and lower 10th or 25th percentiles could be used, for example.

Whilst developing the methodology described in Sect. 3, the geostatistics analysis route known as 'Kriging', or Gaussian process regression, was also tested (Bailey and Gatrel, 1995; Lophaven et al., 2002). This analysis produces the best linear unbiased estimator of the actual underlying TCO at times intermediate to the observations, and also produces an associated uncertainty. In brief, it performed well for days where there are a larger number of contributing observations, but showed poorer performance during winter or other days with few observations. This latter issue is in part due to the complex nature of applying the method, where for few observations there is a risk of overfitting. However for studies where short-term prediction of the TCO and its near term uncertainty is of interest, such as real-time estimates or nowcasting, Kriging may find applications. More generally its applications could include spatial analysis and interpolation of TCO and surface irradiance, two fields where global datasets are reliant on a limited number of measurement sites.

## 6 Conclusions

In this study we propose, describe and assess a new methodology for determining a more representative best daily value of total column ozone from Brewer spectrophotometer observations. This method overcomes the limitations of making the traditional choice between a possibly small number of direct sun measurements and zenith sky measurements. It requires a homogenised set of DS and ZS data as a pre-requisite, but then, by taking a weighted mean and accounting for both the uncertainty associated with each individual observation and the time period the observation represents, produces a more representative value based on the full set of daily observations.

Applying the new method to the 2000–2016 dataset from Brewer 172 stationed at Manchester [53.47° N, 2.23° W], we show that the number of contributing observations is more than doubled from an average of 10.72 to 24.06 per day; increased

numbers of observations are found in both summer and winter, though the fractional increase is greater during the summer. Similarly the interdecile range of mean representative times is approximately halved throughout the year, whilst the time span of contributing observations is skewed towards longer hours, predominantly during the summer months. Together these findings demonstrates that the method results in a substantial improvement in sampling and utilisation of valid observations,

and hence improves the representativeness of the daily mean TCO. The issue of rejecting otherwise valid data is also removed. We find no evidence of impact on supra-annual trends from applying the new method, though the ground-satellite bias is increased for this station by 2.8 DU. We also note that a change in daily sampling when one instrument type replaces another at a site could contribute to introduce a small step in the data record, and similarly, care should be taken if reprocessing only a partial data record.

To complement our proposed BRDV calculation, we also recommend reporting the standard error of the daily mean value, and replacing the standard deviation by a more complete measure of the daily spread such as the upper and lower limits of the interdecile range, or simply the maximum and minimum observed values.

**Data availability**

The underlying data used in this study can be accessed at the World Ozone and Ultra-Violet Data Centre (Smedley et al., 2017).

**Author contribution**

A.R.D. Smedley was primarily responsible for the data collection, processing and monitoring of Brewer spectrophotometer #172 and led the manuscript preparation. J.S. Rimmer assisted with data collection, contributed to the manuscript and securing

of funding. A.R. Webb contributed to the manuscript, securing of funding and was Principal Investigator on the overall grants.

**Acknowledgements**

Stratospheric ozone and spectral UV baseline monitoring in the United Kingdom is supported by DEFRA, The Department for the Environment, Food, and Rural Affairs, since 2003.

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

**Table 1. Summary statistics for data shown in figure 2: number of contributing observations, representative observation time, and** time span of contributing observations. For each case values shown are arithmetic means, and in brackets, lower 10th percentile and upper 10th percentile.

|  | No. of Observations | Rep. time [h] | Time span [h] |
|---|---|---|---|
| Annual (trad.) | 10.72 [ 2 24] | 12.12 [10.19 14.24] | 5.15 [ 0.06 10.94] |
| Annual (new) | 24.06 [ 6 46] | 12.02 [11.01 13.02] | 7.61 [ 2.50 12.35] |
| Summer [MJJ] (trad.) | 14.68 [ 2 31] | 12.22 [ 9.26 15.56] | 7.76 [ 0.50 12.25] |
| Summer [MJJ] (new) | 36.68 [21 55] | 11.99 [10.63 13.33] | 11.56 [10.00 13.00] |
| Winter [NDJ] (trad.) | 5.71 [ 1 11] | 12.00 [11.27 12.83] | 2.18 [ 0.00 4.00] |
| Winter [NDJ] (new) | 8.54 [ 4 16] | 12.01 [11.40 12.60] | 2.92 [ 1.50 4.50] |

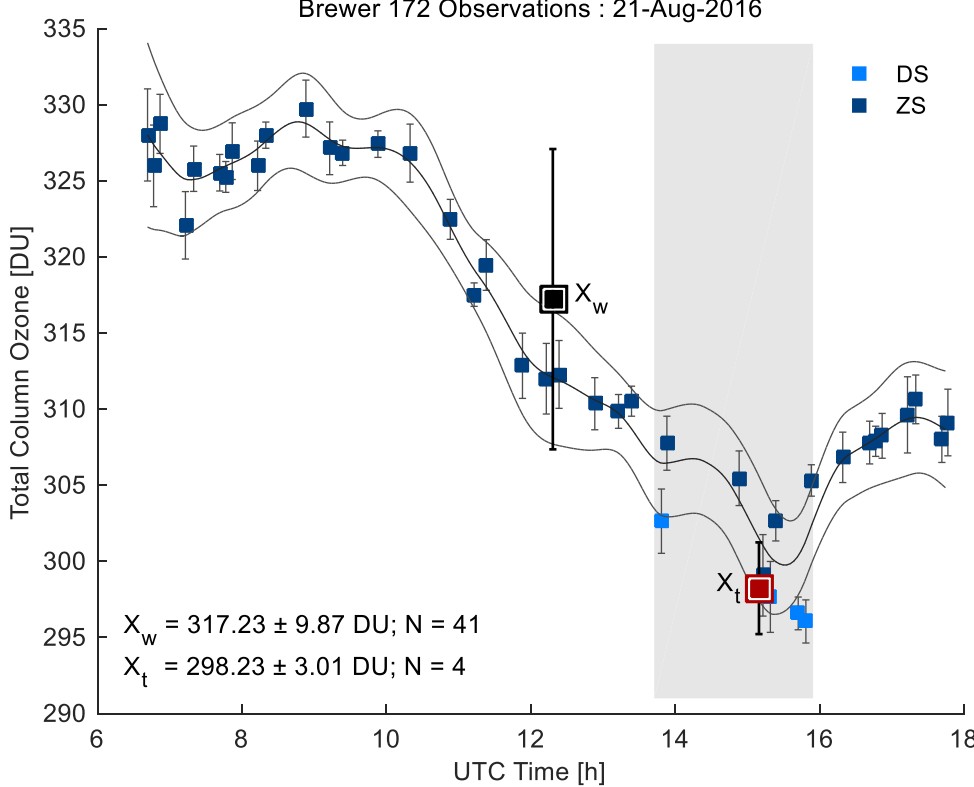

**Figure 1. Example day showing valid DS and ZS observations and their standard deviations. Also shown are the daily representative value based on the traditional (arithmetic mean, DS>ZS preference) methodology (red outlined square, $X_t$), and the daily representative value formed through the method described herein (black outlined square, $X_w$). The shaded area shows the time coverage for data points contributing to the traditional estimate of the BRDV, and N is the number of contributing observations to each BRDV estimate.**

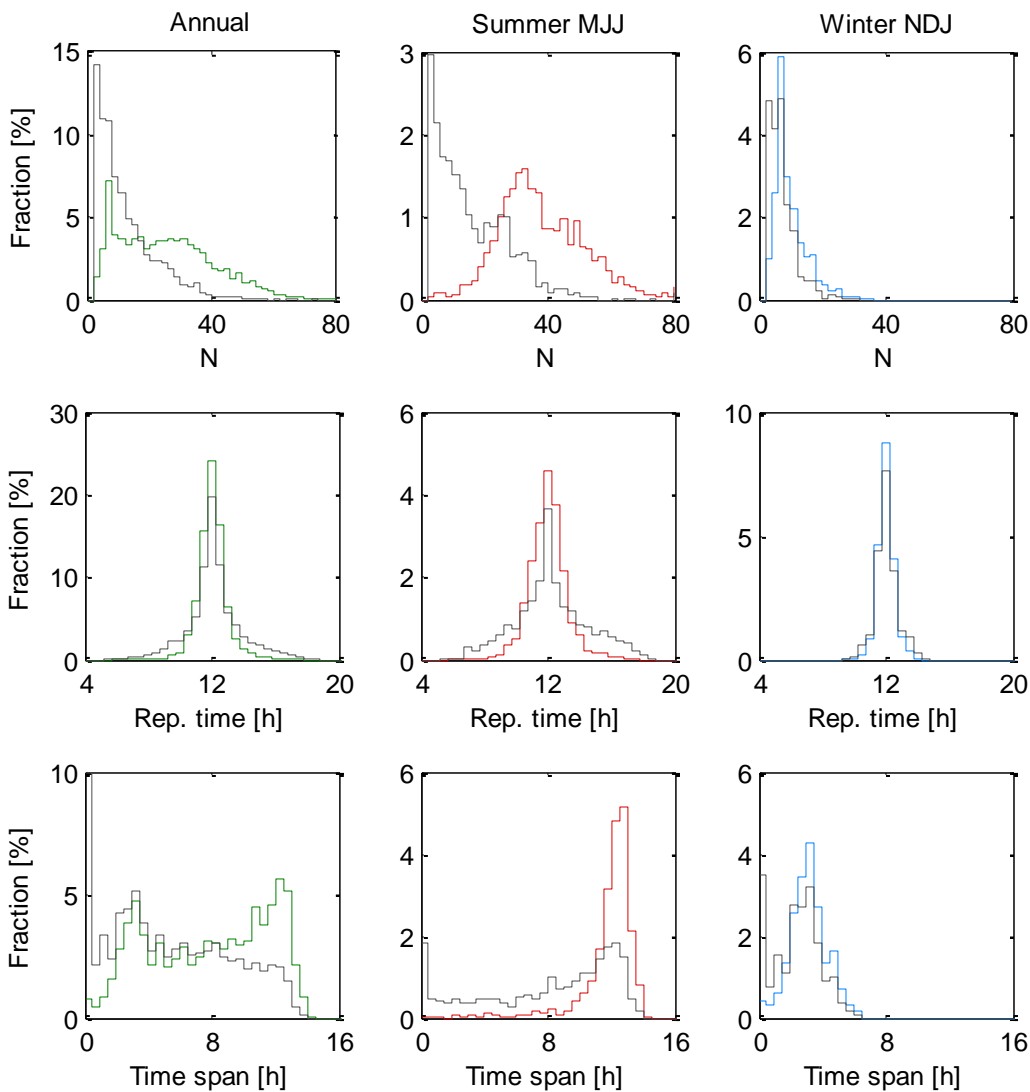

**Figure 2. Histograms of number of contributing observations (N, first row), representative observation time (second row), and time span of contributing observations (third row) for 2000–2016 all months (first column), summer months (May-Jun-Jul, second column), and winter months (Nov-Dec-Jan, third column). Grey traces show results from traditional method, coloured traces show results for methodology described in the present study.**

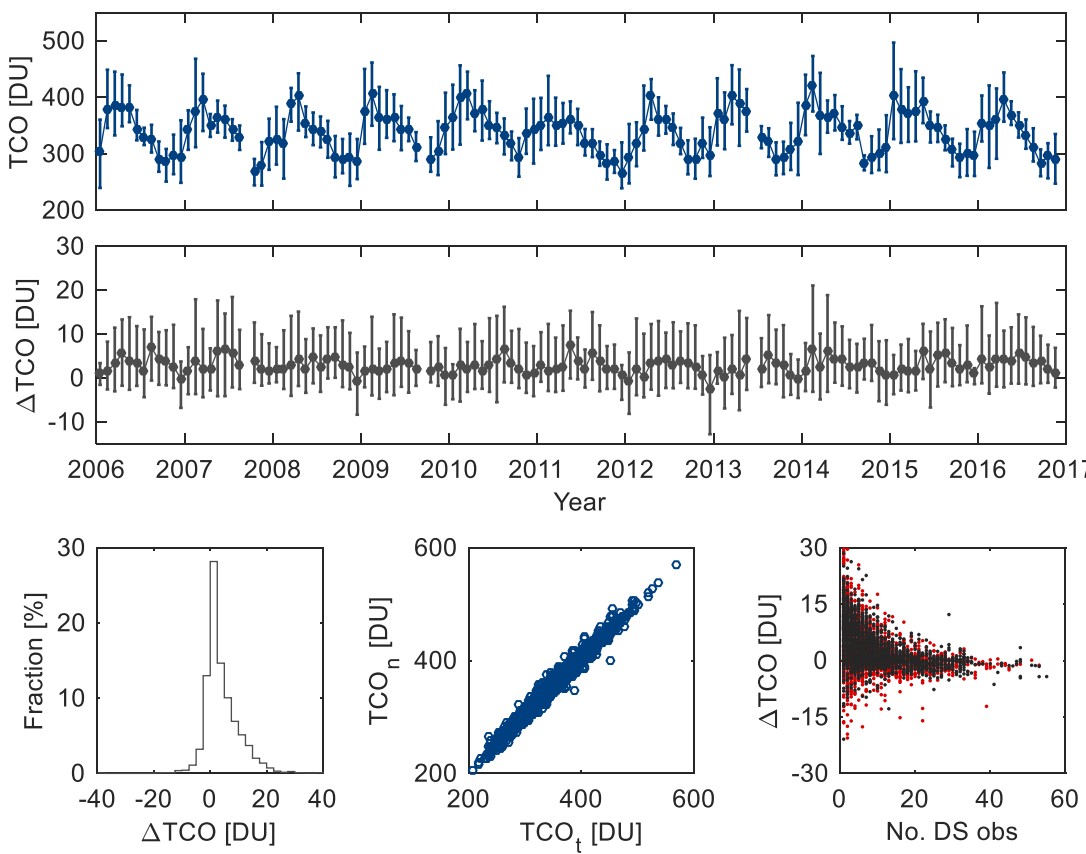

**Figure 3. Upper full width panel: Traditional monthly mean TCO; line length shows upper and lower monthly 10th percentiles. Lower full width panel: Mean monthly daily difference between new and traditional best representative values, plus upper and lower monthly 10th percentiles. Lower left panel: Histogram of daily differences between traditional and new daily TCO calculations. Lower middle panel: Scatterplot of new daily TCO values against traditional TCO. Lower right panel: Scatterplot of daily difference between new and traditional BRDV against number of contributing DS measurements; dark grey markers for data points from months JJASON, red markers from remainder.**

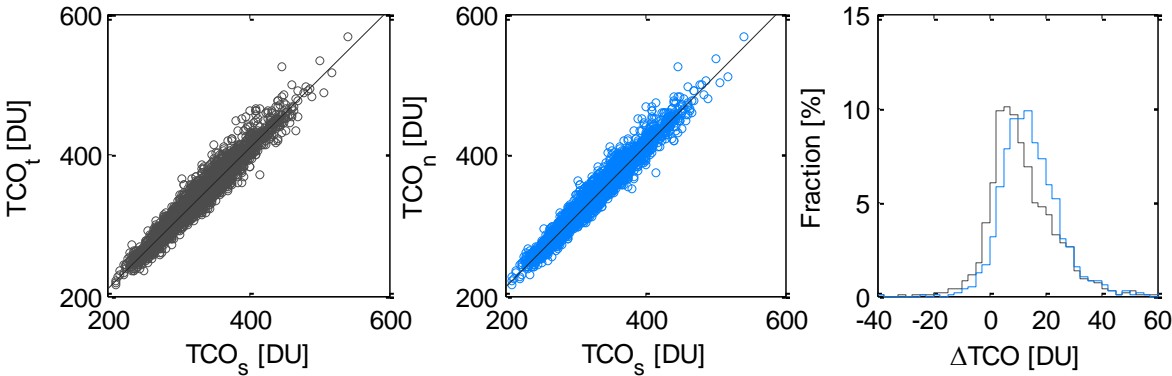

**Figure 4. Left panel: Traditional BRDV with traditional DS-ZS choice vs OMI mean overpass. Middle panel: BRDV from described methodology vs OMI mean overpass. Right panel: Histogram of daily differences between traditional (grey) and new (blue) daily TCO calculations vs satellite overpass TCO.**

