# Peer review of "A more representative "best representative value" for daily total column ozone reporting"

_Atmospheric Measurement Techniques, 2017_

## Referee Comment (RC1) · Anonymous Referee #2 · 2 Aug 2017

The manuscript presents a new methodology to calculate the best representative daily value (BRDV) of total ozone column measured with Brewer spectrophotometers by using a combination of direct sun and zenith sky observations. The new methodology will clearly have an advantage at some observation sites, because significant ozone variations on a time scale of hours in combination with a limited number of clear sky conditions are better represented with the new method. However, it needs a very good characterisation of the instrument in order to establish the relation between direct sun and zenith sky observations with a small uncertainty.

In general, I think one has to have in mind also, what the purpose of the reported BRDV is. If it is used for validation of satellite estimates of total ozone, then it is more important to focus on the time of the overpass of the satellite. That could reduce the standard

deviation of the results, although when using a large number of observations then the average might be unchanged. On p.6, In. 9-19, this aspect is already discussed, although not shown in detail because 'the results were very similar' to the daily mean values. If the BRDV is used to calculate UV exposure, then the value of the ozone column around local noon would be most significant. Furthermore, the 'daily mean' is anyway biased systematically by the different day length throughout the year, whereas the ozone variations on an hourly time scale are in general independent of daylight or night time.

The manuscript is very well written and clearly structured with clear figures. I suggest only one specific point to be clarified by the authors prior to final publication:

p.5, In. 29: the mean difference between old and new method constitutes 2.79 DU. This should be discussed a bit more detailed as it seems to be significant, due the very high number of samples. It is stated that this is in the order of the 'calibration uncertainty'; does this mean the calibration of the zenith sky relative to the direct sun observations has this significant bias? On the example day shown in Fig. 1 one can also see such a bias (even considering the standard deviations of the individual measurements). A similar bias is also found in the comparison with satellite data (p. 6, In. 12/13).

Overall I think the manuscript is worthwhile to be published in AMT, as it will stimulate an important discussion in the Brewer community. I suggest publication after minor revision.

---

## Referee Comment (RC2) · V. Savastiouk (Referee) · 9 Aug 2017

The comment was uploaded in the form of a supplement:
https://www.atmos-meas-tech-discuss.net/amt-2017-181/amt-2017-181-RC2-supplement.pdf

---

## Referee Comment (RC3) · Anonymous Referee #3 · 5 Sep 2017

In the manuscript, the authors describe and assess a new methodology for determining a best representative daily value (BRDV) of total column ozone from Brewer spectrophotometer observations. The authors propose a method, which take into account the possibility of changes in the total ozone column (TOC) during the day, and having unevenly sampled observations. All measurements (DS and ZS) of the day are taken into account in case they meet the specified validity criteria and have passed the tail removal check. This is a welcome approach, especially at sites, where cloudiness is frequent and the number of DS measurement is restricted. It would be pity not to use good ZS measurements, as it would be the case if choosing the "traditional" way, where possibly only couple of day's DS measurements would be used. The paper is well written and easy to follow, and is a good opening for discussion in the ozone measurement

community. I recommend the paper to be published after minor corrections. Please find here below my specific comments.

Specific comments:

- Please add as reference Karppinen et al. 2016, (especially Figures 2 and 5). Citation: Karppinen, T., Lakkala, K., Karhu, J. M., Heikkinen, P., Kivi, R., and Kyrö, E.: Brewer spectrometer total ozone column measurements in Sodankylä, Geosci. Instrum. Method. Data Syst., 5, 229-239, https://doi.org/10.5194/gi-5-229-2016, 2016.

- Page 3, line 1: not five wavelengths for ozone measurements?

- Chapter 3, Figure 1: Could there be a bias/offset between the DS and ZS measurements of around 5 DU? Looking at quasi simultaneous DS and ZS measurements at around 14h and 16h, there seems to be a difference of around 5 DU, can you explain this?

- FIG 1: For days like this, also for cases with evenly spaced DS measurement and changing TOC during the day: why reporting the daily mean? Why not reporting all measurements?

- Could also other type of measurements be included in the analyze (FZ, FM)?

-Page 3, line 31: What is the range of biases between ds and zs observations?

-Page 4: increasing the limit value for ZS to 4.0: Does it increase the uncertainty of the measurement?

-Page 4, line 6: Include a reference to differences in stray light rejection of single and double Brewers.

-page 4, line 20: How did you end up to choose, for the first (and last) data point, to use the length of the first(last) inter-observation time period? Why not the half of it? What if you have one good ZS measurement in the early morning, no measurements during late morning/midday and many data points (e.g. DS) in the afternoon, isn't it

that in your method the ZS from the morning will get a big weight as the time t1 in eq. 2. will be big? Can it cause problems, thinking that ZS are not as "good" as DS measurements?

-Page 5, What is the min SZA (airmass) in Manchester?

-Page 5: Refer earlier to FIG 2 (actually I didn't find any), otherwise it is difficult to follow the discussion at lines 11-19.

-Page 5: I don't find the definition of "representative time", how did you calculate it?

-Page 6: satellite comparison: Is the satellite TOC data meant to represent the daily mean TOC, or is it the overpass time TOC? I don't really see the point to compare the satellite data with other than the nearest of the overpass time. And the satellite should be compared to ground based, not ground-based to satellite. I don't really think that the bias between ground-based and satellite data is due to "that ground-validation of satellites relies upon the traditional methodology", as it can be concluded from your sentence at line 14. But maybe from the way the satellite algorithm is built, satellite instrument errors/uncertainties, etc.

-Page 6: Chapter 5, please add discussion about the problem that e.g. in the morning, the Brewer is looking to East at low solar elevation, and in the afternoon/evening to West at low solar elevation: The geographical location, to which the TOC is calculated, is not really above the measurement station. In the morning at East from the station, in the Evening at West from the station. What if there is a strong strong East-West ozone gradient? Could the min air mass measurement be after all the most representative for the specific site ?

Technical corrections

-Missing references in the Reference list: Kerr et al., 1981 Smedley et al., 2010

- Page 3, line 17: "recorded between 1348 and 1549" ->between 13:48 - 15:49 UTC?

- Figure 1: Time, UTC ?

- Table 1: Explain in the caption what is rep. time (even if we can see it from Figure 2).
* * *

---

## Author Comment (AC1) · 20 Oct 2017

Our response to RC1 Referee #2 can be found at pp1-2 in the supplement pdf

Please also note the supplement to this comment:
https://www.atmos-meas-tech-discuss.net/amt-2017-181/amt-2017-181-AC1-
supplement.pdf

---

## Author Comment (AC2) · 20 Oct 2017

Our response to RC2 Referee #1 can be found at pp2-3 in the supplement pdf

Please also note the supplement to this comment:
https://www.atmos-meas-tech-discuss.net/amt-2017-181/amt-2017-181-AC2-supplement.pdf

---

## Author Comment (AC3) · 20 Oct 2017

Our response to RC3 Referee #3 can be found at pp3-6 in the supplement pdf

Please also note the supplement to this comment:
https://www.atmos-meas-tech-discuss.net/amt-2017-181/amt-2017-181-AC3-supplement.pdf

---

## Author Response (AR1)

**Response to referees**

We thank all three referees for their comments, and their recognition that the present study should open a discussion on improved calculations of the best representative daily value. We have expanded and modified the manuscript in response to all three referees.

Our response to their combined comments is structured below with the original comment in *italic* text, our response in normal text, noting where changes will be made in the revised manuscript with underlined text.

**RC1: Anonymous Referee #2**

*The manuscript presents a new methodology to calculate the best representative daily value (BRDV) of total ozone column measured with Brewer spectrophotometers by using a combination of direct sun and zenith sky observations. The new methodology will clearly have an advantage at some observation sites, because significant ozone variations on a time scale of hours in combination with a limited number of clear sky conditions are better represented with the new method. However, it needs a very good characterisation of the instrument in order to establish the relation between direct sun and zenith sky observations with a small uncertainty.*

We agree with the referee that to merge DS and ZS values in this way relies on both measurement types being well-characterised and having small uncertainties. Much effort has been put into improving direct sun measurements and calibrations over the years, with less emphasis on the zenith sky measurement mode. However it should be borne in mind that zenith sky measurements are dependent on the DS mode calibration, and as such, improvements in the determination of these values, and overall instrument stability should lead to improvements in both measurement modes. As a simple safeguard we take the approach of re-determining the ZS polynomials for each inter-calibration period (p3, ll29-30), and in any event, we note that there is the option of applying expanded ZS uncertainties to the relative weightings (p4, ll 29-30).

*In general, I think one has to have in mind also, what the purpose of the reported BRDV is. If it is used for validation of satellite estimates of total ozone, then it is more important to focus on the time of the overpass of the satellite. That could reduce the standard deviation of the results, although when using a large number of observations then the average might be unchanged. On p.6, ln. 9-19, this aspect is already discussed, although not shown in detail because 'the results were very similar' to the daily mean values. If the BRDV is used to calculate UV exposure, then the value of the ozone column around local noon would be most significant. Furthermore, the 'daily mean' is anyway biased systematically by the different day length throughout the year, whereas the ozone variations on an hourly time scale are in general independent of daylight or night time.*

Our approach was to try and determine a daily value based on the fullest possible set of DS and ZS data available (so to include all valid data, rather than rejecting a potentially large portion). This proposed technique reduces bias away from noon, and so would be a more appropriate measure for daily UV exposure and a more representative value for time series investigations than the current calculation. It is intended as a proposed improvement over the current daily mean value submitted to WOUDC, and users of these values. The referee is correct in that for satellite overpass validation, or high time resolution analysis, individual observations are likely a better choice. Though even here, some choice between DS and ZS measurements would have to be made.

*The manuscript is very well written and clearly structured with clear figures. I suggest only one specific point to be clarified by the authors prior to final publication:*

*p.5, ln. 29: the mean difference between old and new method constitutes 2.79 DU. This should be discussed a bit more detailed as it seems to be significant, due the very high number of samples. It is stated that this is in the order of the 'calibration uncertainty'; does this mean the calibration of the zenith sky relative to the direct sun observations has this significant bias? On the example day shown*

*in Fig. 1 one can also see such a bias (even considering the standard deviations of the individual measurements). A similar bias is also found in the comparison with satellite data (p. 6, ln. 12/13).*

There are two issues at play here. The ZS polynomials are calculated as a single fit over all cloud conditions present at the site during each inter-calibration period. i.e. considering the mean sky and cloud conditions (overall the fitting procedure gives a bias of only -0.4DU, with a spread between individual DS-ZS pairs of 6.3DU). For a specific day we expect some small DS-ZS biases due to the particular cloud conditions present and those that the referee notes in figure 1 are consistent with the spread of residual DS-ZS values (now noted at p3, ll31-32). The bias of 2.79DU found between traditional and proposed methods, and also exhibited in the comparison to satellite data, instead relates to the new method's increased daily sampling. On this point there are two potential contributions: increased sampling over longer daylight hours where perhaps the TCO is increased, and increased reliance on different internal filters at low solar elevations. In the middle panel of figure 3 there is a suggestion that greater differences are seen during summer when the time span is increased the most (see also figure 2, third row). As to our comment that the value of 2.79DU was of the order of the calibration uncertainty – we referred to the typical spread of values measured by different instruments immediately after calibration (0.5 to 1.0%).

p6, ll24-29: We have expanded and clarified these points in the text.

*Overall I think the manuscript is worthwhile to be published in AMT, as it will stimulate an important discussion in the Brewer community. I suggest publication after minor revision.*

**RC2: Referee #1**

*This paper is well written and raises a very important question of the daily mean total ozone reporting protocol from the ground-based network of the Brewer spectrophotometers with implications to other type of ground-based instruments. The paper is well written and easy to follow.*

*The proposed method for calculating the best representative value for daily total column ozone (BRV) using a weighted combination of the direct-sun and the zenith-sky observations will improve the "representative" part of the reported values.*

*The paper correctly states that the ground-based observations are used in a variety of ways and need to serve the entire ozone monitoring community in the best possible way.*

*p.1 says that reporting to WOUDC "has traditionally been predicated upon a binomial choice between direct sun and zenith sky observations". This is not quite accurate – there are codes available for many observation types including the focused moon and the ozone values derived from the global irradiance measurements. In fact, WOUDC offers a simple way to register new observation types and report those. Please address this and maybe show how your proposed method can include more than 2 observation types (seems straight forward with weightings).*

We thank referee 1 for raising this point. In the current manuscript we concentrate on DS and ZS measurements, as for Manchester and similar mid-latitude sites they are likely to be those relied upon most often. It is true that the methodology can be extended to combining valid observations from any measurement mode, including focussed moon and global irradiance derived total column ozone values. In general we expect that this could be incorporated into WOUDC submissions most simply by registering a new observation type (weighted mean). It would also require guidelines on how it should be derived (for the sake of consistency) with submitters detailing their procedure in the Scientific Support Statement. Alternatively, if stations submit their raw data, or processed individual observations, the weighted mean BRDV calculation could be applied across all sites as a daily summary value.

p1 l9: "binomial" has been altered to "simple". We note that other observation types are possible at p2 ll5-9.

p5 ll4-11: We have added a paragraph noting that the method could be extended to other observation types and its practical implementation in terms of submissions to WOUDC.

*The results of applying the proposed method shows some differences compared to the "traditional" method. The paper does provide some comments, but this may require a more detailed discussion. For example, is there a correlation between more ZS only data and high ozone variability during a day that may produce significantly different daily mean depending on when the observations took place?*

In an earlier draft we planned to include a figure and more discussion on a closely related point, but we focussed on investigating a correlation between the daily TCO variability and the BRDV differences. There was no clear relation and so it was not included in the submitted version. However we have revisited this aspect, concentrating on the absolute number of contributing DS observations and using the time of year as a coarse proxy for daily variability. There is a clear relation between these parameters: a much greater potential change in the BRDV is exhibited when the number of DS observations nears unity (in line with our assertion in the introduction that partly cloudy days with a small number of DS observations may be most misrepresented). The seasonal effect is fairly weak.

p7 l1-8: We have included an additional panel in figure 3 and discussion at this point in the text.

p1 ll15-16: A sentence has been added to the abstract on this point

*The paper makes a good point of suggesting some improvements to the data reporting. What about submission od the individual data points rather than daily averages? Would this not be better for all purposes? WOUDC has a format for this in place.*

We agree with referee #1 that reporting individual observations is an option, and would have benefits for certain classes of analysis (satellite validation or detailed studies, for example). For long time series analysis where a consistent approach is needed, or for users who are less well-versed in the details of Brewer operation and data analysis, our view is that a summary value for each day is of benefit to the wider community (see also first response to RC2 on practical implementations).

*With minor revision/expansion this paper should be published and will provide a good argument to start a larger discussion on the data submission from the Brewers.*

**RC3: Anonymous Referee #3**

*In the manuscript, the authors describe and assess a new methodology for determining a best representative daily value (BRDV) of total column ozone from Brewer spectrophotometer observations. The authors propose a method, which take into account the possibility of changes in the total ozone column (TOC) during the day, and having unevenly sampled observations. All measurements (DS and ZS) of the day are taken into account in case they meet the specified validity criteria and have passed the tail removal check. This is a welcome approach, especially at sites, where cloudiness is frequent and the number of DS measurement is restricted. It would be pity not to use good ZS measurements, as it would be the case if choosing the "traditional" way, where possibly only couple of day's DS measurements would be used. The paper is well written and easy to follow, and is a good opening for discussion in the ozone measurement community. I recommend the paper to be published after minor corrections. Please find here below my specific comments.*

*Specific comments:*

*- Please add as reference Karppinen et al. 2016, (especially Figures 2 and 5). Citation: Karppinen, T., Lakkala, K., Karhu, J. M., Heikkinen, P., Kivi, R., and Kyrö, E.: Brewer spectrometer total ozone column measurements in Sodankylä, Geosci. Instrum. Method. Data Syst., 5, 229-239, https://doi.org/10.5194/gi-5-229-2016, 2016.*

p5, l6: Karppinen et al (2016) is now referred to in a brief discussion of extending the methodology to other observation types.

*- Page 3, line 1: not five wavelengths for ozone measurements?*

Five operational wavelengths, plus a dark channel, are used during the instrument observation, though in the standard analysis only four of these operational wavelengths are processed to a final value. We have clarified the text on this point.

p3 l1: The number of wavelengths used has been clarified.

*- Chapter 3, Figure 1: Could there be a bias/offset between the DS and ZS measurements of around 5 DU? Looking at quasi simultaneous DS and ZS measurements at around 14h and 16h, there seems to be a difference of around 5 DU, can you explain this?*

We expect, and noted, small biases between DS and ZS observations on a particular day. This is attributed to the particular cloud conditions present; the ZS polynomial determination is a single fit over all cloud conditions present at the site during each inter-calibration period. It is for this reason we suggest that one option is to expand the ZS uncertainties at the end of p4. This could be done on a daily basis for days where there are valid DS and ZS measurements.

p4 l30 to p5 l2: We have added the phrase "day-to-day" and an explanatory phrase on this point.

*- FIG 1: For days like this, also for cases with evenly spaced DS measurement and changing TOC during the day: why reporting the daily mean? Why not reporting all measurements?*

Reporting all the valid measurements is one option. However to provide a consistently determined summary value for each day has benefits for time series analysis or users who wish to know the ozone for secondary analysis. Whether the daily mean value should be provided by users, or by WOUDC by processing of individual measurements is a point for discussion among the community. The former is closer to the current system, but of course the latter would provide a more consistent approach.

*- Could also other type of measurements be included in the analyze (FZ, FM)?*

Yes, see our first response to RC1.

*- Page 3, line 31: What is the range of biases between ds and zs observations?*

The overall DS–ZS bias is minimal at –0.4DU. The standard deviation of the population of DS-ZS pairs is 6.3DU.

p3 ll31-32: We have added details of the bias for our dataset here.

*- Page 4: increasing the limit value for ZS to 4.0: Does it increase the uncertainty of the measurement?*

There will be a small increase in the uncertainty of the BRDV, but this outweighed by the benefit of an increased number of data points and sampling frequency.

*- Page 4, line 6: Include a reference to differences in stray light rejection of single and double Brewers.*

p4, l7: We have added a reference at the relevant point that notes the differences in stray light rejection between single and double instruments

*- Page 4, line 20: How did you end up to choose, for the first (and last) data point, to use the length of the first(last) inter-observation time period? Why not the half of it?*

The time weighting for all other data points is from the midpoint of the preceding inter-observation time interval to the midpoint of the following inter-observation time interval. For equally spaced observations, this weighting is equal to the inter-observation time interval. Therefore for the first (and last) data point we chose to weight this equivalently.

*What if you have one good ZS measurement in the early morning, no measurements during late morning/midday and many data points (e.g. DS) in the afternoon, isn't it that in your method the ZS*

*from the morning will get a big weight as the time t1 in eq. 2. will be big? Can it cause problems, thinking that ZS are not as "good" as DS measurements?*

It is possible to construct sequences of measurements that seemingly cause problems such as this, yes. However, provided care has been taken to determine up to date ZS polynomials, and if necessary, expand the ZS uncertainty, then the authors do not think that the proposed methodology provides a biased result. If in the scenario described there were a strong gradient of ozone, then it would be better to include the morning ZS measurement (provided it is valid and hence has a small uncertainty), than not, when determining the best representative daily value. The referee is correct of course that the long time interval between the ZS and first DS would result in a larger weighting, but this would be offset if it had a larger uncertainty. The first DS will also be given a larger weighting due to the break in valid observations. When there is such a gap in observations, the question is what knowledge do we have about the ozone column during this period? It was this point that lead us to explore the potential benefits of 'Kriging', which would provide a best estimate (and uncertainties) during the gap, based only on the surrounding observations. For many days' observations we obtained good fits, but on occasion spurious results were returned. On balance the weighting proposed was a simpler and more robust route to a BRDV.

*- Page 5, What is the min SZA (airmass) in Manchester?*

The minimum airmass observed in Manchester is approximately 1.15.

p5 l27-28: We have added the information at this point.

*- Page 5: Refer earlier to FIG 2 (actually I didn't find any), otherwise it is difficult to follow the discussion at lines 11-19.*

Figure 2 and table 1 now referred to earlier.

p5 l27: Reference to figure 2 added.

*- Page 5: I don't find the definition of "representative time", how did you calculate it?*

We follow the standard Brewer output, defining representative time as the mean of valid observation times weighted by their TCO values.

P6 ll4-5. This explanation is now added to the text here.

*- Page 6: satellite comparison: Is the satellite TOC data meant to represent the daily mean TOC, or is it the overpass time TOC? I don't really see the point to compare the satellite data with other than the nearest of the overpass time. And the satellite should be compared to ground based, not ground-based to satellite. I don't really think that the bias between ground-based and satellite data is due to "that ground-validation of satellites relies upon the traditional methodology", as it can be concluded from your sentence at line 14. But maybe from the way the satellite algorithm is built, satellite instrument errors/uncertainties, etc.*

The referee is correct in that it is usually best to compare the satellite TCO to ground-based TCO measured at the overpass time. Here though our aim was to use the satellite record as a common reference to explore how moving from the traditional method to the proposed method of calculating daily means might affect future comparisons between the two instrument sets. In the first part of this comparison we therefore use the mean satellite TCO where there is more than one overpass per day; in the latter part we focussed on selecting DS/ZS measurements close to the satellite overpass time.

p7 l19: added "mean" to clarify satellite data

p6 l22: Line containing quote has been removed

*- Page 6: Chapter 5, please add discussion about the problem that e.g. in the morning, the Brewer is looking to East at low solar elevation, and in the afternoon/evening to West at low solar elevation: The*

*geographical location, to which the TOC is calculated, is not really above the measurement station. In the morning at East from the station, in the Evening at West from the station. What if there is a strong strong East-West ozone gradient? Could the min air mass measurement be after all the most representative for the specific site?*

This is an issue that affects any calculation of a daily value, and impacts comparison between ground-based and satellite-retrieved datasets. However, the authors are not persuaded that even in these particular circumstances that the minimum airmass measurement is necessarily an improvement over the proposed method. First if there was a stationary linear gradient as the referee suggests, with e.g. 280DU measured at 9am, 300DU at noon, and 320DU at 3pm, then with equally spaced observations, both the minimum airmass and the suggested method would yield a value of 300DU. If observations were not equally spaced, due to cloud cover, then the traditional method would give a bias, and a minimum airmass measurement may not be available. If the gradient was not linear, then the minimum airmass measurement would be biased according to the overall conditions present through the day. In any event additionally specifying the maximum and minimum would aid in summarising the conditions present during the day, as we mention in Sect. 5.

p5 l13-23: At this point in the text we add a discussion on the problem of sampling along the slant path and how traditional and proposed methods, as well as the minimum airmass measurement, fare.

*Technical corrections*

*- Missing references in the Reference list: Kerr et al., 1981 Smedley et al., 2010*

p2 l30: Smedley et al year corrected

p10 ll10-13: Kerr et al reference added.

*- Page 3, line 17: "recorded between 1348 and 1549" ->between 13:48 - 15:49 UTC?*

p3 l17: Time format corrected.

*- Figure 1: Time, UTC ?*

p11, fig 1. X-axis label clarified

*- Table 1: Explain in the caption what is rep. time (even if we can see it from Figure 2).*

p10, table 1: Revised caption now includes full description of "rep. time" and other statistics.

[revised manuscript text omitted]